# Clusters of longitudinal risk profile trajectories are associated with cardiometabolic diseases: Results from the population-based KORA cohort

**Fiona Niedermayer**[1,2,3], **Gunther Schauberger**[3], **Wolfgang Rathmann**[4,5], **Stefanie J. Klug**[3], **Barbara Thorand**[2,4], **Annette Peters**[1,2,4,6], **Susanne Rospleszcz**[1,2,6,7] *

**1** Chair of Epidemiology, IBE, Faculty of Medicine, LMU Munich, Munich, Germany, **2** Institute of Epidemiology, Helmholtz Zentrum München, German Research Center for Environmental Health, Neuherberg, Germany, **3** Chair of Epidemiology, Department of Sport and Health Sciences, Technical University of Munich, Munich, Germany, **4** German Center for Diabetes Research (DZD), München-Neuherberg, Neuherberg, Germany, **5** Department for Biometrics and Epidemiology, German Diabetes Research Institute, Leibniz Institute for Diabetes Research at Heinrich Heine University, Düsseldorf, Germany, **6** German Center for Cardiovascular Disease Research (DZHK), Munich Heart Alliance, Munich, Germany, **7** Department of Diagnostic and Interventional Radiology, University Medical Center Freiburg, Faculty of Medicine, University of Freiburg, Freiburg, Germany

* Susanne.rospleszcz@helmholtz-muenchen.de

## Abstract

### Background

Multiple risk factors contribute jointly to the development and progression of cardiometabolic diseases. Therefore, joint longitudinal trajectories of multiple risk factors might represent different degrees of cardiometabolic risk.

### Methods

We analyzed population-based data comprising three examinations (Exam 1: 1999–2001, Exam 2: 2006–2008, Exam 3: 2013–2014) of 976 male and 1004 female participants of the KORA cohort (Southern Germany). Participants were followed up for cardiometabolic diseases, including cardiovascular mortality, myocardial infarction and stroke, or a diagnosis of type 2 diabetes, until 2016. Longitudinal multivariate k-means clustering identified sex-specific trajectory clusters based on nine cardiometabolic risk factors (age, systolic and diastolic blood pressure, body-mass-index, waist circumference, Hemoglobin-A1c, total cholesterol, high- and low-density lipoprotein cholesterol). Associations between clusters and cardiometabolic events were assessed by logistic regression models.

### Results

We identified three trajectory clusters for men and women, respectively. Trajectory clusters reflected a distinct distribution of cardiometabolic risk burden and were associated with prevalent cardiometabolic disease at Exam 3 (men: odds ratio (OR)$_{ClusterII}$ = 2.0, 95% confidence interval: (0.9–4.5); OR$_{ClusterIII}$ = 10.5 (4.8–22.9); women: OR$_{ClusterII}$ = 1.7 (0.6–

**Data Availability Statement:** The datasets analyzed during the current study are not publicly available due national data protection laws, since the informed consent given by KORA study

participants does not cover data posting in public databases. Data are available upon request by means of a project agreement from KORA. Requests should be sent to kora.passt@helmholtz-muenchen.de and are subject to approval by the KORA Board. Analysis codes are available from the authors upon reasonable request.

**Funding:** This project has been financed in part through HGF Future Topic AMPro. The KORA study was initiated and financed by the Helmholtz Zentrum München – German Research Center for Environmental Health, which is funded by the German Federal Ministry of Education and Research (BMBF) and by the State of Bavaria. Furthermore, KORA research was supported within the Munich Center of Health Sciences (MC-Health), Ludwig-Maximilians-Universität, as part of LMUinnovativ. The funders had no role in study design, data collection and analysis, decision to publish, or preparation of the manuscript.

**Competing interests:** The authors have declared that no competing interests exist.

**Abbreviations:** ANOVA, Analysis of Variance; BMI, Body-Mass-Index; BP, Blood Pressure; CI, Confidence Interval; CVD, Cardiovascular diseases; FRS, Framingham Risk Score; HDL, High-density lipoprotein; HR, Hazard Ratio; KORA, Cooperative Health Research of Augsburg; LDL, Low-density lipoprotein; OR, Odds Ratio; T2D, Type 2 diabetes.

4.7); $OR_{ClusterIII}$ = 5.8 (2.6–12.9)). Trajectory clusters were furthermore associated with incident cardiometabolic cases after Exam 3 (men: $OR_{ClusterII}$ = 3.5 (1.1–15.6); $OR_{ClusterIII}$ = 7.5 (2.4–32.7); women: $OR_{ClusterII}$ = 5.0 (1.1–34.1); $OR_{ClusterIII}$ = 8.0 (2.2–51.7)). Associations remained significant after adjusting for a single time point cardiovascular risk score (Framingham).

## Conclusions

On a population-based level, distinct longitudinal risk profiles over a 14-year time period are differentially associated with cardiometabolic events. Our results suggest that longitudinal data may provide additional information beyond single time-point measures. Their inclusion in cardiometabolic risk assessment might improve early identification of individuals at risk.

## Introduction

Cardiometabolic diseases, such as cardiovascular diseases (CVD) and type 2 diabetes (T2D), are among the leading causes of mortality and morbidity worldwide. Today, these diseases represent a significant global burden on the healthcare system [1, 2].

Development of cardiometabolic diseases is multifaceted and risk factors include obesity, physical inactivity, dietary factors, dyslipidemia and hypertension, with relative risk factor contributions being different in men and women [3, 4]. Most of these risk factors are modifiable and can be targeted by primordial or primary prevention. Therefore, identification of high-risk individuals at an early stage is crucial for timely intervention and risk factor treatment. Risk assessment is usually done with established risk scores, such as the Framingham Risk Score [5] or the Finnish or German Diabetes Risk Score [6, 7], which incorporate multiple risk factors. These scores only include measures from a single time point. However, individual cardiometabolic risk builds up long-term and can change over time [8, 9]. Consequently, joint modelling of multiple risk factors allows to evaluate how risk factors longitudinally interact and influence each other.

Longitudinal trajectories convey information about the individual progression and risk development. This has been recognized in recent studies that investigated associations of blood pressure or lipid changes with CVD [10–12]. Increasing cumulative 10-year systolic blood pressure levels were associated with higher lifetime risk of CVD [10] and incident CVD [11]. In the Framingham Heart Study, long-term lipid levels over approximately 30 years were more strongly associated with elevated coronary artery calcium than single time-point lipid levels [12]. Clustering, i.e. grouping together individuals with similar trajectories, can be used to identify patterns [13] and also allow for even earlier identification of at-risk individuals beyond the already known cut-off values [14, 15].

First associations between model-based trajectory clusters and cardiometabolic events have already been demonstrated [16–20]. However, trajectory clusters based on a single risk factor cannot account for the multifactorial etiology of cardiometabolic diseases. To evaluate the interplay of multiple risk factors, Niiranen et al. [21] calculated a multivariate trajectory score based on each single identified risk factor trajectory cluster and found an association with a 2.8-fold higher risk of incident CVD. Apart from this work, reports on multivariate trajectory clusters are scarce–yet these clusters would be able to account for both the multifactorial etiology and the temporal risk factor evolution of cardiometabolic diseases. Furthermore,

women develop CVD on average later in life than men [22], which cannot be attributed solely to biological differences, but may be due to the lack of sex-specific cut-offs for cardiometabolic markers leading to delayed diagnosis and initiation of treatment [3, 4]. In this regard, few studies have considered possible sex-specific clustering of cardiometabolic risk factors [19, 21, 23].

To address this issue, the present study aims to identify distinct clusters of sex-specific risk profile trajectories over a 14-year time horizon on a population-based level and evaluate their association with prevalent and incident cardiometabolic events.

## Materials and methods

### Study sample

We used longitudinal data from the KORA ("Cooperative Health Research in the Region Augsburg") S4 sample, which is a population-based cohort study from Bavaria, Germany. A detailed description of recruitment, sampling, and examination protocols is presented elsewhere [24]. The original cohort was repeatedly examined at three time points (Exam 1, Exam 2, Exam 3). Briefly, 4,261 individuals participated in the baseline assessment (Exam 1, S4, 1999–2001), 3,080 of these participated in the seven-year follow-up (Exam 2, F4, 2006–2008) and 2,279 individuals of the original sample participated in the 14-year follow-up (Exam 3, FF4, 2013–2014). After Exam 3, participants were followed up until 2016 for mortality, and morbidity of several chronic diseases.

All KORA studies comply with the Declaration of Helsinki. Each participant gave written informed consent and ethical approval was granted by the Ethics Committee of the Bavarian Medical Association and the Bavarian commissioner for data protection and privacy [S4/Exam 1: EC No. 99186, F4/Exams 2 and FF4/Exam 3: EC No. 06068.]. For the current analysis, data access was granted on April 15, 2019. Authors had no means to access information that could identify individual participants during or after data collection.

For the present analysis, we used a final analytical sample of 1980 individuals with complete data until the end of follow-up 2016. We excluded 118 participants who missed Exam 2 and 181 participants were excluded because they had missing data in any of the risk factors of interest. A detailed participant flow is presented in S1 Fig in S1 File.

### Risk factor assessment

Trained staff performed standardized physical examinations, blood draw and face-to-face interviews at all three examinations. A detailed description of measurement methods is described elsewhere [25–27]. Briefly, low-density lipoprotein (LDL) cholesterol, high-density lipoprotein (HDL) cholesterol and total cholesterol were measured by enzymatic photometric assays at Exam 1 and by enzymatic colorimetric Flex assays at Exam 2 and 3. Hemoglobin—A1c (HbA1c) was assessed by a turbidimetric inhibition immunoassay at Exam 1 and by a cation-exchange high performance liquid chromatographic photometric assay at Exam 2 and 3. More details are provided in S1 Table in S1 File.

Height and weight were measured by Seca's measuring systems (Seca GmbH & Co, KG, Hamburg). Body-Mass Index (BMI) was calculated as weight divided by squared height (kg/m$^2$). Waist circumference (WC) was determined by using an inelastic tape at the level midway between the lower rib margin and the iliac crest.

Systolic and diastolic blood pressure (BP) were measured on the right arm after the participant rested for at least five minutes. An OMRON type HEM-705CP device was used at all examinations. The measurement was repeated three times with a three-minute interval in-between, and the mean of the second and third measurement was used as the final value.

Medication intake, physical activity, smoking behavior and alcohol consumption were self-reported, as described before [28–30]. Participants were categorized into never-smoker, ex-smoker, or smoker and the number of pack years were calculated. Physical activity was estimated based on participants reported duration of leisure times spent on sport activities based on four possible answers: (1) >2 hours per week, (2) 1–2 hours per week, (3) <1 hour per week and (4) none, separately for winter and summer. A total score was obtained by summing the numbers (1)-(4) relating to summer and winter. Participants were classified as physically active if their total score was < 5; otherwise, they were classified as "physically inactive" [29]. The Framingham Risk score was calculated according to the original formula at all three examinations [5].

## Outcome assessment

Outcomes of interest were twofold. 1) Any first cardiometabolic event defined as cardiovascular mortality, nonfatal and fatal myocardial infarction and stroke, or a diagnosis of T2D after Exam 1, but before, or at, Exam 3. These were evaluated as prevalent events at Exam 3. 2) Any first incident cardiometabolic event after Exam 3 until the end of follow up in 2016. These were evaluated as incident events after Exam 3. Data on causes of deaths were obtained from official death certificates, collected by health departments, physicians, or coroners. Mortality due to any cardiovascular event was defined by the ICD-9 codes (390–459; 798). Myocardial infarction of individuals aged below 75 were registered by the Augsburg Hospital Myocardial Registry. Data on stroke or myocardial infarction of participants aged older than 75 or of participants moving out of the study area were identified by self-report assessed in regular follow-up questionnaires or telephone interviews. All self-reported events were validated by contacting treating physicians or by sighting health records. T2D was defined as a self-reported diagnosis by a physician or self-reported intake of antidiabetic medication and again validated by contacting treating physicians or by sighting health records.

## Statistical analysis

The statistical analysis workflow is presented in S2 Fig in S1 File. Briefly, after descriptive statistics of the study sample, we derived longitudinal trajectory clusters, described the distribution of events and event-free survival according to clusters, and quantified the association of clusters with prevalent and incident cardiometabolic events.

**Descriptive statistics.** All analyses were stratified by sex. Individual risk factors at each examination are presented as means and standard deviations for continuous variables and as counts and percentages for categorical variables. For skewed variables, medians are additionally provided. Changes over time were assessed with repeated measures ANOVA (skewed variables: Friedman test) and Cochran's Q test, respectively. To assess differences between excluded individuals and individuals from the final sample, their risk factor values at Exam 1 were compared by t-test, Wilcoxon-Test or $X^2$-Test, where applicable.

**Identification of longitudinal trajectory clusters.** We used non-parametric k-means with Euclidean distance to identify sex-specific clusters of longitudinal risk factor trajectories. Since the distance calculation with categorical variables using Euclidean distance is not straightforward, we only considered continuous variables for clustering. Nine established cardiometabolic risk factors, obtained at all three examinations, were included: age, HDL cholesterol, LDL cholesterol, total cholesterol, BMI, WC, HbA1c, systolic BP and diastolic BP. Selection of these variables was based on their established role in cardiometabolic disease and their ready availability in clinical assessment. In principle, k-means groups subjects with similar trajectories, i.e., risk factor values at each Exam and similar changes over time, into a

cluster. Individuals with dissimilar trajectories are assigned to different clusters. (Dis-)
similarity was determined by Euclidean distance.

Since k-means is sensitive to outliers [31], all risk factor data except age were winsorized
at the sex-specific 95%-quantile. Standardization per trajectory and subsequent clustering
were performed with the R package "kml3d" [32]. The final number of clusters was
determined by Calinski-Harabasz criterion while allowing up to eight clusters (S2 Table in
S1 File). Mean risk factor levels at each exam according to cluster membership were
visualized by line plots.

**Association of trajectory clusters with cardiometabolic events.** Distribution of
cardiometabolic events according to cluster membership was visualized by bar plots. To assess
the association of trajectory clusters with the first outcome of interest–incident events after
Exam 1, but before, or at Exam 3, and thus prevalent at Exam 3—logistic regression models
adjusted for age at baseline were calculated. Additionally, Kaplan-Meier curves were used to
visualize how the cumulative incidence of first cardiometabolic events during the 14-year
horizon between Exam 1 and Exam 3 evolved according to cluster membership. The sample
used for visualization with Kaplan-Meier curves was identical to the sample where we
analyzed the first outcome of interest, except for exclusion of n = 72 prevalent cases at baseline
(S2 Fig, S4 Table in S1 File). Differences according to cluster membership were evaluated by
log-rank test.

To assess the association of trajectory clusters with the second outcome of interest—
incident events after Exam 3 -, all prevalent diseases at Exam 3 (n = 292, S2 Fig and S4 Table in
S1 File) were excluded. Logistic regression models then were calculated and adjusted for the
Framingham Risk Score at Exam 3, in order to examine whether cluster membership could
provide additional information over the score.

As sensitivity analyses, CVD and T2D events were considered separately. Results of the
logistic regression are given as Odds Ratios (OR), corresponding 95%- confidence intervals
(CI) and p-values. A two-sided p-value <0.05 was considered to indicate statistical
significance. All statistical analyses were performed with R 4.0.5 (R Core Team, Vienna,
Austria).

## Results

### Study sample

The final sample comprised 976 men and 1004 women. At baseline, men were on average 48
years and women 47 years old. During the 14-year horizon between Exam 1 and Exam 3, all
risk factors except triglycerides in women significantly changed. Mean blood pressure levels
were declining, while intake of antihypertensive medication increased, and mean HDL
cholesterol levels were u-shaped for both men and women. Mean BMI and WC increased in
both sexes. While men had decreasing mean LDL cholesterol and total cholesterol levels, these
lipid levels increased or remained stable in women. Intake of lipid-lowering medication
increased in both sexes. Mean HbA1c levels significantly rose in men but declined in women.
At Exam 3, more participants were physically active and fewer smoked (Table 1).

Compared to participants retained in the final analysis, excluded participants had a worse
cardiometabolic profile at baseline (S3 Table in S1 File). On average, these participants were
older, had higher mean levels in systolic blood pressure, BMI, WC, blood lipids, C-reactive
protein and were less physically active. In addition, more subjects were taking
antihypertensive medications, and more subjects were diagnosed with T2D or experienced a
stroke before Exam 1.

**Table 1. Characteristics of male and female participants at Exam 1, Exam 2 and Exam 3.**

| | Men (n = 976) | | | | Women (n = 1004) | | | |
|---|---|---|---|---|---|---|---|---|
| | Exam 1 (1999–2001) | Exam 2 (2006–2008) | Exam 3 (2013–2014) | p-value | Exam 1 (1999–2001) | Exam 2 (2006–2008) | Exam 3 (2013–2014) | p-value |
| Age [years] | 47.5 ± 12.6 | 54.7 ± 12.5 | 61.1 ± 12.5 | | 46.9 ± 12.0 | 54.1 ± 12.0 | 60.5 ± 12.0 | |
| *Blood pressure (BP)* | | | | | | | | |
| Systolic BP [mmHg] | 132.6 ± 16.4 | 127.4 ± 17.0 | 124.2 ± 16.4 | <0.001 | 120.6 ± 17.7 | 115.7 ± 16.7 | 114.2 ± 17.2 | <0.001 |
| Diastolic BP [mmHg] | 83.2 ± 10.1 | 78.1 ± 9.6 | 75.1 ± 10.1 | <0.001 | 77.5 ± 9.9 | 72.9 ± 9.2 | 70.7 ± 8.8 | <0.001 |
| *Anthropometric factors* | | | | | | | | |
| Body-Mass Index [kg/m$^2$] | 27.3 ± 3.8 | 27.8 ± 4.1 | 28.2 ± 4.4 | <0.001 | 26.4 ± 4.9 | 26.9 ± 5.1 | 27.5 ± 5.4 | <0.001 |
| Weight [kg] | 84.3 ± 12.3 | 86.5 ± 14.0 | 87.1 ± 15.0 | <0.001 | 69.6 ± 13.0 | 71.1 ± 13.5 | 72.0 ± 14.4 | <0.001 |
| Waist circumference [cm] | 96.4 ± 10.2 | 99.2 ± 12.0 | 102.8 ± 12.1 | <0.001 | 83.4 ± 11.7 | 86.8 ± 12.9 | 91.3 ± 13.6 | <0.001 |
| *Lipids* | | | | | | | | |
| Total cholesterol [mg/dl] | 226.3 ± 41.4 | 213.1 ± 37.0 | 210.5 ± 39.1 | <0.001 | 223.2 ± 40.6 | 217.8 ± 39.3 | 223.3 ± 38.9 | <0.001 |
| HDL cholesterol [mg/dl] | 51.3 ± 13.5 | 50.2 ± 11.8 | 58.1 ± 15.5 | <0.001 | 65.3 ± 17.4 | 62.2 ± 14.4 | 73.2 ± 18.5 | <0.001 |
| LDL cholesterol [mg/dl] | 139.7 ± 39.4 | 137.6 ± 32.6 | 134.0 ± 34.9 | <0.001 | 129.5 ± 39.7 | 133.9 ± 35.5 | 136.3 ± 36.0 | <0.001 |
| Triglycerides [mg/dl][ab] | 157.4 ± 105.7 Median: 129.5 | 141.3 ± 104.9 Median: 118.0 | 135.9 ± 79.7 Median: 115.7 | <0.001 | 120.4 ± 65.3 Median: 106.0 | 100.9 ± 55.6 Median: 88.0 | 108.4 ± 51.9 Median: 96.8 | 0.292 |
| *Glucose metabolism* | | | | | | | | |
| HbA1c [%] | 5.5 ± 0.6 | 5.5 ± 0.6 | 5.6 ± 0.7 | 0.014 | 5.6 ± 0.5 | 5.5 ± 0.5 | 5.5 ± 0.6 | <0.001 |
| 2-hour glucose [mg/dl][a,b] | 120.0 ± 49.4 Median: 108.0 | 111.9 ± 35.9 Median: 100.0 | 116.7 ± 42.5 Median: 107.0 | <0.001 | 119.1 ± 44.5 Median: 109.0 | 107.1 ± 35.8 Median:100.0 | 109.8 ± 39.8 Median: 101.0 | <0.001 |
| Fasting blood glucose [mg/dl][a] | 103.6 ± 16.1 | 100.9 ± 18.2 | 106.5 ± 22.6 | <0.001 | 98.4 ± 14.3 | 93.2 ± 13.9 | 99.2 ± 20.4 | <0.001 |
| *Inflammation* | | | | | | | | |
| C-reactive protein [mg/L][b] | 2.0 ± 3.6 Median: 0.9 | 1.9 ± 3.0 Median: 1.0 | 2.2 ± 3.7 Median: 1.1 | <0.001 | 2.7 ± 5.0 Median: 1.2 | 2.3 ± 4.0 Median: 1.1 | 2.6 ± 5.1 Median: 1.3 | <0.001 |
| *Medication* | | | | | | | | |
| Antihypertensive | 126 (12.9%) | 271 (27.8%) | 364 (37.3%) | <0.001 | 126 (12.6%) | 232 (23.1%) | 350 (34.9%) | <0.001 |
| Lipid-lowering | 57 (5.8%) | 121 (12.4%) | 188 (19.3%) | <0.001 | 42 (4.2%) | 95 (9.5%) | 142 (14.1%) | <0.001 |
| *CVD risk* | | | | | | | | |
| Framingham risk score [%] | 13.7 ± 12.4 | 17.3 ± 14.5 | 19.3 ± 14.4 | <0.001 | 5.6 ± 6.0 | 6.7 ± 6.8 | 7.9 ± 7.6 | <0.001 |
| *Lifestyle factors* | | | | | | | | |
| Alcohol consumption [g/day][b] | 23.7 ± 24.7 Median: 17.1 | 21.9 ± 24.2 Median: 17.0 | 21.7 ± 24.0 Median: 15.8 | 0.003 | 8.8 ± 12.0 Median: 2.9 | 8.1 ± 12.1 Median: 2.9 | 8.5 ± 12.8 Median: 2.9 | <0.001 |
| Pack years[b] | 14.7 ± 21.7 Median: 5.9 | 16.3 ± 23.4 Median: 6.6 | 17.1 ± 24.3 Median: 7.8 | <0.001 | 6.0 ± 10.2 Median: 0.0 | 6.8 ± 11.5 Median: 0.0 | 7.3 ± 12.3 Median: 0.0 | <0.001 |
| Physically active | 518 (53.1%) | 558 (57.2%) | 550 (56.4%) | 0.030 | 541 (53.9%) | 597 (59.5%) | 594 (59.2%) | <0.001 |
| Smoking | | | | <0.001 | | | | <0.001 |
| never-smoker | 333 (34.1%) | 333 (34.1%) | 333 (34.1%) | | 526 (52.4%) | 524 (52.2%) | 521 (51.9%) | |
| ex-smoker | 399 (40.9%) | 459 (47.0%) | 486 (49.8%) | | 274 (27.3%) | 328 (32.7%) | 347 (34.6%) | |
| smoker | 244 (25.0%) | 184 (18.9%) | 157 (16.1%) | | 204 (20.3%) | 152 (15.1%) | 136 (13.6%) | |

Continuous variables are presented as mean ± standard deviation, p-values are calculated by repeated measures ANOVA indicating whether mean values differ significantly on at least one exam. Categorical variables are presented as counts and percentages, p-values are calculated by Cochran's Q test and indicating whether counts differ significantly on at least one exam.

[a]Sample size deviate (Men: Fasting blood glucose: $N_{Exam1}$ = 293, $N_{Exam2}$ = 971, $N_{Exam3}$ = 965; 2-hour glucose: $N_{Exam1}$ = 291, $N_{Exam2}$: 896, $N_{Exam3}$ = 831; triglycerides: $N_{Exam1}$ = 360; Women: Fasting blood glucose: $N_{Exam1}$ = 268, $N_{Exam2}$ = 997, $N_{Exam3}$ = 990; 2-hour glucose: $N_{Exam1}$ = 260, $N_{Exam2}$: 942, $N_{Exam3}$ = 874; triglycerides: $N_{Exam1}$ = 327).

[b]Friedman-Test was used due to skewed distribution

## Characterization of longitudinal trajectory clusters

Based on Calinski-Harabasz criterion, the optimal number of longitudinal risk profile trajectory clusters was three for both men and women (S2 Table in S1 File). These clusters differed in mean risk factor levels and changes over time.

For men, Cluster I comprised 310 subjects (32% of the male sample). Subjects were on average the youngest at baseline (39.7 ± 11.3 years), had the lowest mean levels of BMI, waist circumference, systolic blood pressure and HbA1c and the highest mean levels of HDL cholesterol over the entire study period (Fig 1 and S5 Table in S1 File). Mean BMI ranged from normal weight to borderline overweight (Exam 1: 24.5 ± 2.2 kg/m$^2$; Exam 3: 25.4 ± 2.5 kg/m$^2$) according to the WHO guidelines [33]. Mean systolic blood pressure was elevated to normal and mean diastolic blood pressure was normal (<80 mm/Hg) according to the American Heart Association blood pressure guidelines [34]. Subjects had on average increasing LDL cholesterol levels (Exam 1: 108.1 ± 25.7 mg/dl; Exam 3: 122.7 ± 26.2 mg/dl) within an optimal lipid range according to the Adult Treatment Panel guidelines for treatment of high blood cholesterol [35]. Total cholesterol levels (Exam 1: 193.1 ± 27.5 mg/dl; Exam 3: 200.3 ± 29.8 mg/dl) increased from normal to borderline high but were still the lowest at Exam 1 and Exam 2 compared to the other clusters. Cluster II included 345 participants (35% of the male sample). Individuals had highest mean values of LDL cholesterol and total cholesterol at all examinations ranging from borderline high to high lipid levels (LDL cholesterol: 130–189 mg/dl, total cholesterol: 200 - ≥ 240 mg/dl [35]). In addition, mean values of HDL cholesterol, HbA1c, BMI, waist circumference, systolic blood pressure and diastolic blood pressure at baseline resided between Cluster I and Cluster III. At Exam 3, mean diastolic blood pressure (76.4 ± 9.1 mmHg) was the highest for subjects included in this cluster, although still normal, while mean systolic blood pressure ranged from stage 1 hypertension (130–139 mmHg) to elevated (120–129 mmHg) [34]. According to the WHO definition [33], these subjects were on average overweight. Cluster III comprised 320 men (33% of the male sample). Throughout the study period, individuals had the highest mean levels of BMI, waist circumference, HbA1c, systolic blood pressure and the lowest mean levels of HDL cholesterol (Fig 1 and S5 Table in S1 File). Moreover, subjects were on average the oldest (54.3 ± 10.7 years) and had the highest diastolic blood pressure (86.4 ± 9.4 mmHg), defined as stage 1 hypertension, at baseline. Cluster III was also characterized by the second highest mean LDL cholesterol and total cholesterol levels. In addition, individuals in this cluster had the strongest decline of blood pressure, e.g., systolic blood pressure decreased from high to elevated mean levels (Exam 1: 138.6 ± 15.5 mmHg; Exam 3: 126.1 ± 15.5 mmHg), probably due to increased intake of antihypertensive medication. Similarly, LDL cholesterol (Exam 1: 140.3 ± 34.2 md/dl; Exam 3: 116.8 ± 30.0 mg/dl) and total cholesterol levels (Exam 1: 226.6 ± 35.5 mg/dl; Exam 3: 189.9 ± 33.5mg/dl) declined over time from borderline high to optimal levels, accompanied by an increased intake in lipid-lowering medication. Mean BMI was consistently in the obese category (≥ 30 kg/m$^2$). In conclusion, in men, Cluster I reflected the lowest, Cluster II medium and Cluster III the highest risk factor burden.

For women, Cluster I comprised 371 subjects (37% of the female sample). Individuals had the lowest mean levels of BMI, waist circumference, systolic and diastolic blood pressure, LDL cholesterol and the highest HDL cholesterol levels throughout study period (Fig 2 and S6 Table in S1 File). Mean HbA1c values decreased to the lowest mean values at Exam 3 (5.3 ± 0.4%). According to established thresholds for obesity [33], hypertension [34] and dyslipidemia [35], subjects had on average normal weight (18.5–25 kg/m$^2$), normal blood pressure (systolic: <120 mmHg; diastolic: <80 mmHg) and near optimal LDL cholesterol levels (100–129 mg/dl), but borderline high mean total cholesterol levels (200–239 mg/dl).

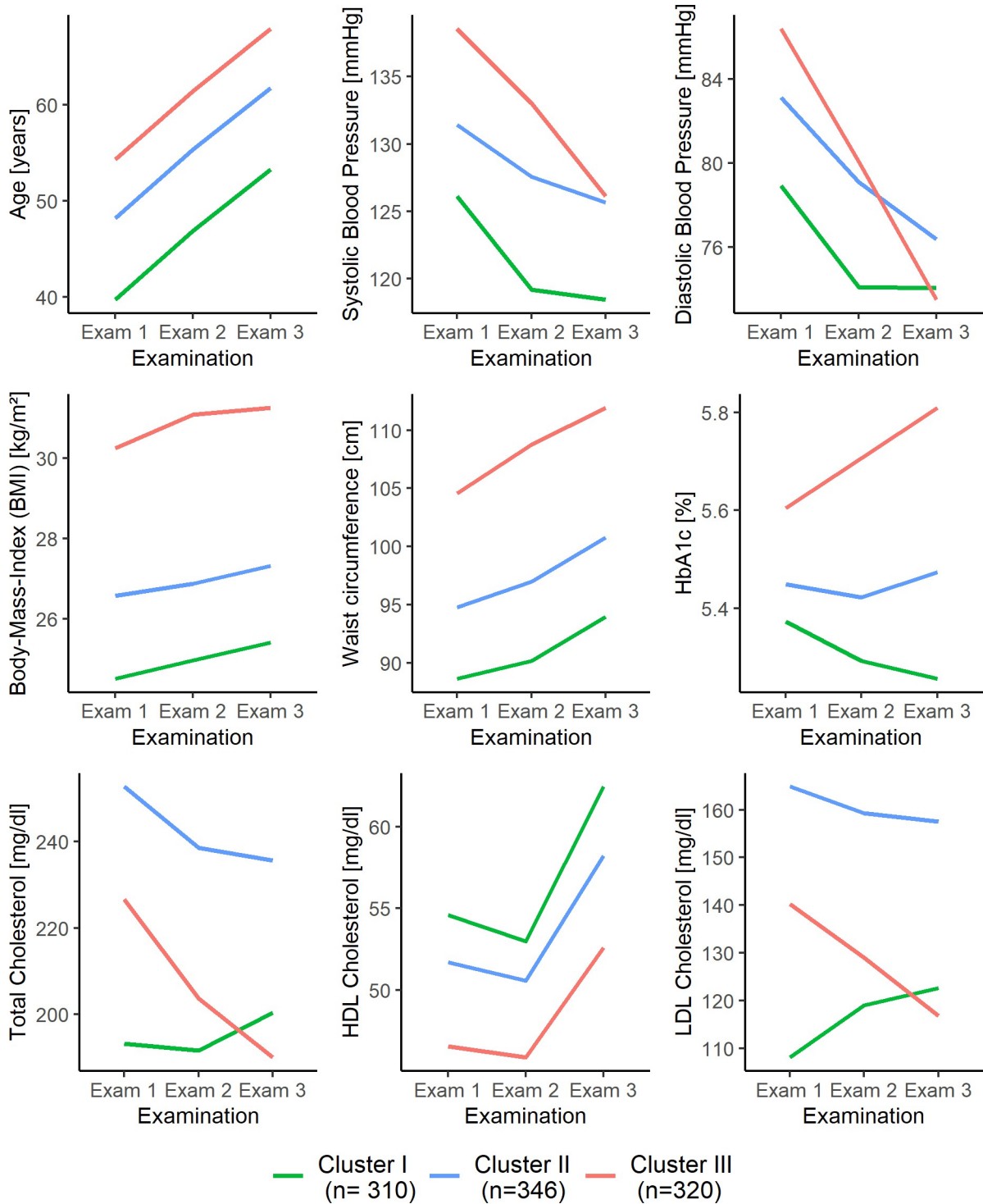

**Fig 1. Risk factor trajectories according to cluster membership of men.** Presented are mean risk factor levels at each examination according to cluster membership. Clusters are based on male participants' longitudinal risk profile trajectories determined by k-means clustering.

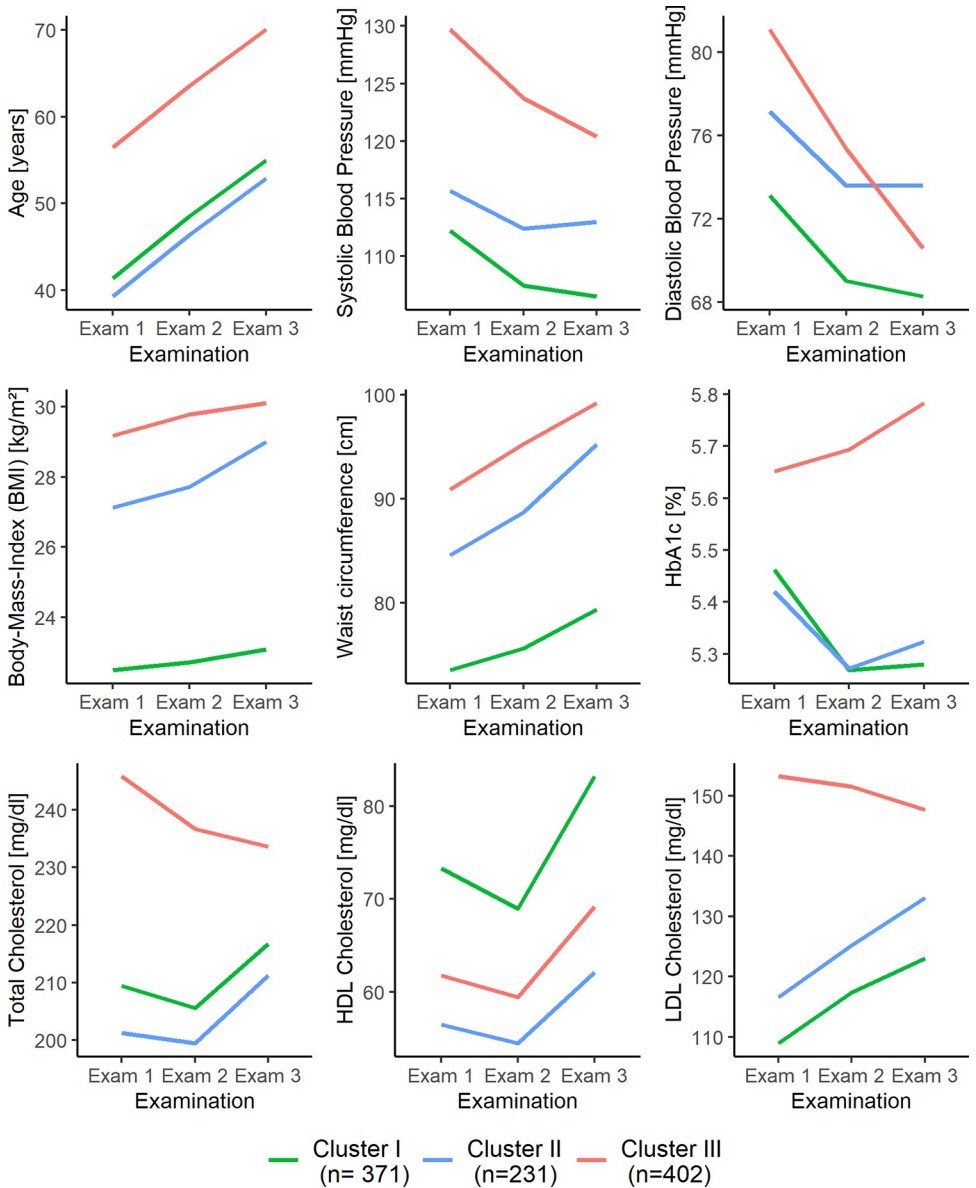

**Fig 2. Risk factor trajectories according to cluster membership of women.** Presented are mean risk factor levels at each examination according to cluster membership. Clusters are based on female participants' longitudinal risk profile trajectories determined by k-means clustering.

Cluster II included 231 subjects (23% of the female sample). Mean levels of BMI, waist circumference, LDL cholesterol and systolic blood pressure resided between Cluster I and Cluster III at every examination (Fig 2 and S6 Table in S1 File). Moreover, subjects were characterized by the lowest mean values of HDL cholesterol (56.5 ± 12.9 mg/dl), total cholesterol (201.3 ± 32.1 mg/dl; borderline high) and were on average the youngest at baseline (39.3 ± 8.0 years). Mean HbA1c was lowest at baseline but slightly increased after Exam 2, resulting in higher mean values at Exam 3 compared to Cluster I. Diastolic blood pressure only decreased between Exam 1 and Exam 2 and then plateaued this level (73.6 ± 7.4 mmHg; normal blood pressure) until Exam 3. Mean BMI was consistently in the overweight category (25.0–29.9 kg/m²). Cluster III comprised 402 participants (40% of the female sample). In

general, participants were characterized by the highest age at baseline (56.5 ± 8.5 years) and by highest mean levels of HbA1c, total cholesterol, LDL cholesterol, BMI, waist circumference and systolic blood pressure over the study period (Fig 2 and S6 Table in S1 File). In contrast to Cluster II, individuals had higher mean HDL cholesterol levels at all examinations. In addition, Cluster III was the only cluster with steadily decreasing mean levels of total cholesterol (Exam 1: 245.8 ± 32.7 mg/dl; Exam 3: 233.6 ± 37.3 mg/dl; range: high to borderline high) and LDL cholesterol (Exam 1: 153.2 ± 30.1 mg/dl; Exam 3: 147.6 ± 33.8; borderline high category) and steadily increasing HbA1c levels over time. The highest decrease in blood pressure was also observed, e.g., mean levels of systolic blood pressure decreased from elevated to normal levels (Exam 1: 129.7 ± 15.4 mmHg; Exam 3: 120.4 ± 15.3 mmHg). In conclusion, in women, Cluster I reflected the lowest, Cluster II medium and Cluster III the highest risk factor burden.

It is important to note that men and women exhibited different behavior in some risk factors, especially for the high-risk Cluster III. While diastolic blood pressure, total cholesterol and LDL cholesterol decreased substantially for men in Cluster III to the lowest mean levels at Exam 3, these levels also decreased for women in Cluster III, but were still at a high-risk level at Exam 3 and highest compared to the other female clusters (Figs 1 and 2). In addition, Cluster II represented decreasing total and LDL cholesterol levels in men but increasing levels in women. The decreasing trend for diastolic blood pressure was stronger in male Cluster II than in female Cluster II. Furthermore, sex differences were also observed for HbA1c in Cluster I and Cluster II.

## Association between trajectory clusters and cardiometabolic events

At Exam 3, there were 170 prevalent cardiometabolic events in men and 112 in women. The majority of events had occurred in Cluster III (Fig 3 and S4 Table in S1 File). As shown in Fig 4, cumulative event-free survival up to Exam 3 was highest in Cluster I and lowest in Cluster III with significant differences between clusters (Log-Rank Test: p<0.001). In logistic regression, for men Cluster II and Cluster III were associated with a 2.7-fold and 16.7-fold higher risk of a prevalent event at Exam 3 compared to Cluster I. This association remained significant only for Cluster III after adjusting for age at Exam 1 (OR = 10.5 (4.8–22.9)). For women, Cluster III was associated with a 11.2-fold higher risk of a prevalent event at Exam 3, which attenuated after adjusting for age at Exam 1 but remained highly statistically significant (OR = 5.8 (2.6–12.9)). For women, Cluster II did not confer a significantly higher risk of a prevalent cardiometabolic event (OR: 1.4 (0.5–4.0)).

After Exam 3, 28 men and 21 women had an incident cardiometabolic event until the end of follow-up. The majority of incident events after Exam 3 occurred in individuals assigned to Cluster III (Fig 3 and S4 Table in S1 File). Cluster III was significantly associated with a higher risk of incident cardiometabolic events in men (OR = 7.5 (95%-CI: 2.4–32.7)) and in women (OR = 8.0 (2.2–51.7)), Table 2. This association attenuated but remained significant after adjusting for the Framingham risk score at Exam 3 (men: $OR_{ClusterIII}$ = 5.2 (1.5–24.0); women: $OR_{ClusterIII}$ = 6.0 (1.4–41.4)).

In sensitivity analyses, we evaluated T2D and CVD separately and found that trajectory clusters were significantly associated with prevalent T2D at Exam 3 for men and women (Table 2, S3 Fig in S1 File). For CVD, Cluster III was associated with a higher risk of prevalent CVD for men, while there was no significant association for women (Table 2, S5 Fig in S1 File). Similar to the main analysis, event-free survival for both T2D and CVD and was highest in Cluster I and lowest in Cluster III (S4 and S6 Figs in S1 File). Log-rank test indicated significant differences across clusters for both T2D and CVD (all p-values <0.001). Due to the

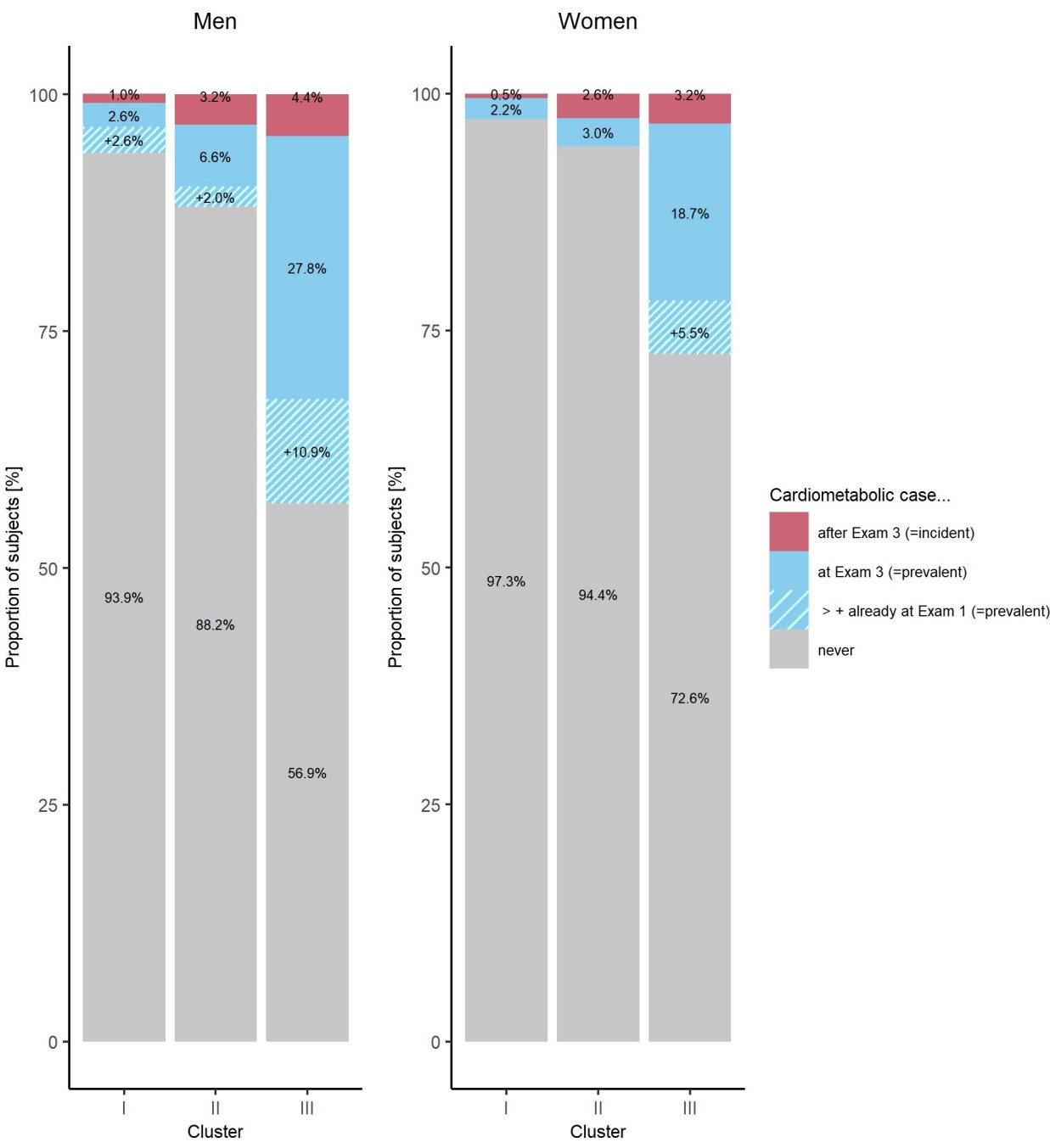

**Fig 3. Distribution of prevalent and incident cardiometabolic cases according to cluster.**

low number of incident events after Exam 3, logistic models were not run separately for CVD and T2D.

## Discussion

We used a non-parametric cluster approach to identify sex-specific cardiometabolic risk profile trajectory clusters in a population-based cohort over a 14-year time period. Our findings are threefold. First, risk profiles significantly changed over time with different mean

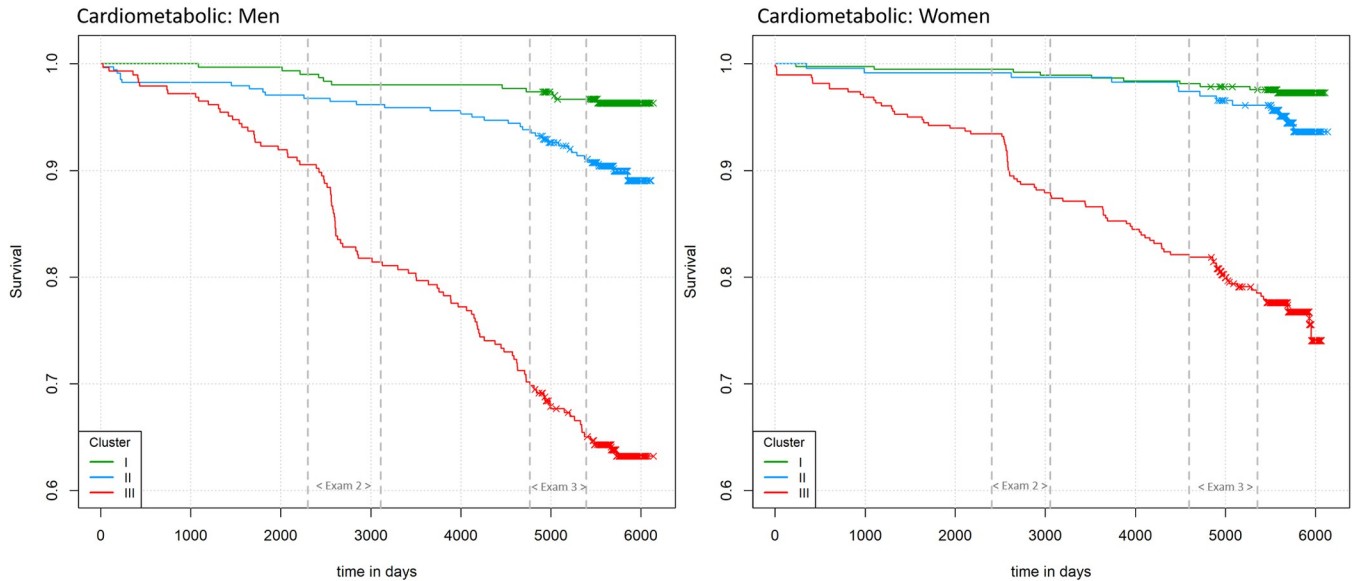

**Fig 4. Kaplan-Meier curves showing cardiometabolic event-free survival stratified by cluster.** On the y-axis: probability of event-free survival without any cardiometabolic event, stratified by cluster membership. On the x-axis: Time from Exam 1 until und of follow-up (3 years after Exam 3) in days. Periods of Exam 2 and 3 are marked by dashed grey lines. Marked decreases during the examination periods are mainly due to newly identified T2D cases that were ascertained during the OGTT performed as part of the examinations. For the visualization of Kaplan-Meier curves, prevalent cardiometabolic events at Exam 1 were excluded, resulting in a sample size of n = 926 for men and n = 982 for women.

**Table 2. Association between cluster membership and cardiometabolic events assessed by logistic regression models.**

| Sex | Outcome | Adjustment Model | Cluster I | Cluster II | | Cluster III | |
|---|---|---|---|---|---|---|---|
| | | | | OR (95%-CI) | p-value | OR (95%-CI) | p-value |
| **Prevalent events at Exam 3** | | | | | | | |
| Men | cardiometabolic | Crude | Reference | 2.7 (1.2–6.1) | 0.019 | 16.7 (7.9–35.2) | <0.001 |
| | cardiometabolic | Crude + Age at Exam 1 | Reference | 2.0 (0.9–4.5) | 0.114 | 10.5 (4.8–22.9) | <0.001 |
| | T2D | Crude | Reference | 2.4 (0.8–6.7) | 0.105 | 19.3 (7.7–48.7) | <0.001 |
| | T2D | Crude + Age at Exam 1 | Reference | 1.7 (0.6–4.8) | 0.344 | 11.4 (4.4–29.8) | <0.001 |
| | CVD | Crude | Reference | 3.3 (0.9–12.1) | 0.066 | 11.7 (3.5–38.9) | <0.001 |
| | CVD | Crude + Age at Exam 1 | Reference | 2.6 (0.7–9.6) | 0.154 | 7.8 (2.2–27.5) | 0.001 |
| Women | cardiometabolic | Crude | Reference | 1.4 (0.5–4.0) | 0.505 | 11.2 (5.3–23.5) | <0.001 |
| | cardiometabolic | Crude + Age at Exam 1 | Reference | 1.7 (0.6–4.7) | 0.332 | 5.8 (2.6–12.9) | <0.001 |
| | T2D | Crude | Reference | 1.6 (0.3–8.1) | 0.560 | 23.0 (7.1–74.0) | <0.001 |
| | T2D | Crude + Age at Exam 1 | Reference | 1.7 (0.3–8.6) | 0.514 | 16.8 (4.9–58.1) | <0.001 |
| | CVD | Crude | Reference | 1.3 (0.3–4.9) | 0.707 | 4.1 (1.5–11.0) | 0.006 |
| | CVD | Crude + Age at Exam 1 | Reference | 2.1 (0.5–8.2) | 0.301 | 1.2 (0.4–3.42) | 0.781 |
| **Incident events after Exam 3** | | | | | | | |
| Men | cardiometabolic | Crude | Reference | 3.5 (1.1–15.6) | 0.056 | 7.5 (2.4–32.7) | 0.002 |
| | cardiometabolic | Crude + FRS at Exam 3 | Reference | 2.6 (0.8–12.1) | 0.161 | 5.2 (1.5–24.0) | 0.017 |
| Women | cardiometabolic | Crude | Reference | 5.0 (1.1–34.1) | 0.051 | 8.0 (2.2–51.7) | 0.006 |
| | cardiometabolic | Crude + FRS at Exam 3 | Reference | 4.7 (1.1–32.6) | 0.058 | 6.0(1.4–41.4) | 0.027 |

Odds Ratios are derived from logistic regression models and denotes the fold-change in risk. For the regression models with prevalent events at Exam 3, prevalent events at Exam 1 were excluded, resulting in a sample size of n = 926 for men and n = 982 for women. For the regression models with incident events after Exam 3, prevalent events at Exam 3 were excluded, resulting in a sample size of n = 806 for men and n = 892 for women. Abbreviations: OR = Odds ratio; CI = Confidence interval; FRS = Framingham Risk Score

lipid trajectories for men and women. Second, we identified three distinct clusters for men and women, which reflected low, moderate, and high cumulative cardiometabolic risk factor burden. Third, trajectory clusters were associated with prevalence of cardiometabolic events at Exam 3 and with incident cardiometabolic events within three years after Exam 3. Furthermore, our results suggest that clusters yielded additional information beyond the Framingham risk score.

Our results thus underscore the clinical importance of considering the longitudinal evolution of multiple risk factors to identify individuals at risk for T2D and CVD early. Cardiometabolic diseases development is multifactorial, and there is substantial interplay between the individual risk factors [36]. However, studies reporting on the joint development of multiple risk factors are scarce. As an exception, Niiranen et al. [21] calculated a multivariate trajectory score based on single risk factor trajectory clusters in the Framingham study. They found a significant association of this score with incident CVD, beyond single-point measures. The importance of longitudinal trajectories is corroborated by further analyses from a subset of the current cohort. Trajectory clusters reflecting a high risk factor burden were associated with worse cardiac function [37] and increased adipose tissue [27], as measured by magnetic resonance imaging. These measures indicate early stages of fully-fledged cardiometabolic diseases. Again, longitudinal trajectories gave additional information beyond single time-point measures [27, 37].

Longitudinal trajectories of single risk factors have already been reported. Previous studies reported mainly increasing blood pressure levels over time [17, 38, 39], in contrast to our findings. This may have different reasons; first, the proportion of subjects on antihypertensive medication might be higher in the KORA cohort compared to other cohorts. Since 2003 systolic blood pressure levels >130 mmHg are classified as prehypertensive [40], potentially leading to earlier treatment of elevated blood pressure. Second, characteristics of participants, especially age and time of enrollment, might differ in other studies. In comparison, studies that reported mainly increasing levels of blood pressure were either conducted earlier [39], when antihypertensive treatment was potentially given only for advanced hypertension, or the sample was younger and therefore, mean blood pressure levels and the number of participants on antihypertensive medications were lower [38].

Additionally, we found the highest risk for cardiometabolic events in Cluster III for men and women. These clusters had the steepest decline of blood pressure and the highest proportion of subjects on antihypertensive medication. This is in good agreement with results from previous studies [17, 41], that demonstrated highest risk of CVD events in clusters with steepest blood pressure decline. Here, the intake of antihypertensive medication seems to represent a proxy for unfavorable blood pressure levels and thus characterize individuals at increased risk.

We observed decreasing levels of lipid parameters in men. This is likely due to increasing number of subjects on lipid-lowering medication. Previous studies provided evidence for this link [42, 43]. So far, three studies identified longitudinal risk factor trajectory clusters of lipid parameters [16, 23, 44]. While trajectories with overall higher HDL cholesterol levels seem to be associated with a favorable cardiovascular profile, trajectories with higher levels of LDL cholesterol or total cholesterol are associated with higher CVD risk. In addition, sex-specific differences in lipid trajectories have already been observed [23]. Recent studies assume this sex difference in lipids can be assigned to sex hormones [45]. Female sex hormones, like estrogens, seem to be protective against cardiovascular diseases and lipid abnormalities. However, this effect is weakened with menopause, which could explain rising blood lipid levels over time [45]. Analogous, we observed increasing total cholesterol and LDL cholesterol levels in Cluster I and Cluster II for women. In these clusters, women were on average 39 and 41 years old at baseline, implying they potentially reached menopause during Exam 2 and Exam 3. However, other explanations for sex-specific trajectories may include differences in the

treatment of CVD risk factors and adherence to therapy. Women are still underrepresented in clinical trials [46–48]; they are more often likely to drop out or discontinue lipid-lowering treatment due to side effects, and physicians are less likely to prescribe statin-based or antihypertensive medications to women than to men [47, 49]. This is supported by our findings, where the reduction of total and LDL cholesterol is higher in men than in women and the proportion of lipid-lowering medication use is higher in men. Moreover, there is still a lack of sex-specific treatment guidelines, e.g., whether target values for CVD risk factors achieved in treatment should be the same for men and women [46, 47].

It is worth noting that we observed stronger effects for T2D compared to CVD in sensitivity analysis, which is likely due to the higher numbers of T2D events. As we included only individuals who participated in all examinations, all fatal CVD cases between the exams were excluded. Since these individuals were naturally at a high risk for CVD, our reported effect of trajectory clusters will underestimate the true effect. Moreover, particularly in women, CVD events might have been underdiagnosed [50]. In contrast, new diabetes cases were diagnosed at each exam by OGTT (S2 Fig in S1 File). In line with this point, participants who dropped out during the study period had a worse cardiometabolic risk profile and had more T2D diagnoses and stroke events at baseline compared to the analytical sample, which may explain the relatively stable cluster trajectories over time.

Our findings are relevant for both clinical and epidemiological applications. The importance of longitudinal data in population-based research on chronic diseases research has been recognized [51]. However, longitudinal data is not only provided in research cohorts, but also in clinical health check-ups and regular visits at physicians, providing a direct practical application in the clinical setting. Using the collected data in a systematic longitudinal approach can inform clinical decision-making and help the physician with early therapeutic and lifestyle intervention for the individual patient. To transfer this into practice, longitudinal data should be included for improving established scores commonly used in clinical settings to identify individuals at risk. This may be particularly relevant for women, since these are at higher risk of not being timely diagnosed and treated due to their different presentation of CVD, like atypical MI symptoms, less inflammatory and fibrotic cardiac markers, and lower prevalence of hypertrophy [22]. On the other side, women tend to have more cardiometabolic disturbances jointly [3, 52, 53]. Therefore, monitoring cardiometabolic risk factors over time allows the identification of sex-specific adverse trajectories and can add valuable information on sex-specific cut-off values for cardiometabolic markers.

Limitations of our study need to be considered. First, although the measurement period with 14 years provided longitudinal data from three repeated examinations, the follow-up was comparatively short with only three years after the last examination. Thus, the number of incident events after Exam 3 was rather low and prevented us from fitting fully adjusted models of incident events, and from conducting separate analyses for incident CVD and T2D. As abovementioned, by including only participants with available data at all examinations, we excluded all who died from CVD in between, which will underestimate the true effect and bias the estimates towards the Null. Finally, we did not conduct any formal modeling whether the cluster membership improve the risk prediction compared to the Framingham risk score. Therefore, the advantage of longitudinal information over single time-point risk scores needs to be further investigated and confirmed in future studies.

## Conclusions

In conclusion, we identified and characterized cardiometabolic longitudinal risk profile trajectories over a 14-year time horizon based on a population-based cohort. Sex-specific risk

profile clusters presented a clear distribution of cardiometabolic risk burden and were associated with cardiometabolic events. Our results emphasize the need of considering longitudinal information in both clinical risk assessment for the individual patient as well as in epidemiological risk prediction models.

## Supporting information

**S1 File.**
(DOCX)

## Acknowledgments

We thank all participants for their long-term commitment to the KORA study, the staff for data collection and research data management and the members of the KORA Study Group (https://www.helmholtz-munich.de/en/epi/cohort/kora) who are responsible for the design and conduct of the study.

## Author Contributions

**Conceptualization:** Fiona Niedermayer, Annette Peters, Susanne Rospleszcz.

**Data curation:** Wolfgang Rathmann, Barbara Thorand, Annette Peters.

**Formal analysis:** Fiona Niedermayer.

**Funding acquisition:** Annette Peters.

**Methodology:** Fiona Niedermayer, Gunther Schauberger, Susanne Rospleszcz.

**Visualization:** Fiona Niedermayer.

**Writing – original draft:** Fiona Niedermayer.

**Writing – review & editing:** Gunther Schauberger, Wolfgang Rathmann, Stefanie J. Klug, Barbara Thorand, Annette Peters, Susanne Rospleszcz.

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
