## [Decision Letter · Decision Letter 0]

16 Jan 2024

PONE-D-23-38330Clusters of longitudinal risk profile trajectories are associated with cardiometabolic diseases: results from the population-based KORA cohortPLOS ONE

Dear Dr. Rospleszcz,

Thank you for submitting your manuscript to PLOS ONE. After careful consideration, we feel that it has merit but does not fully meet PLOS ONE’s publication criteria as it currently stands. Therefore, we invite you to submit a revised version of the manuscript that addresses the points raised during the review process.

We look forward to receiving your revised manuscript.

Kind regards,

Peng Gao, Ph.D.

Academic Editor

PLOS ONE

Journal Requirements:

3. In the online submission form, you indicated that "The datasets analyzed during the current study are not publicly available due national data protection laws, since the informed consent given by KORA study participants does not cover data posting in public databases. Data are available upon request by means of a project agreement from KORA. Requests should be sent to kora.passt@helmholtz-muenchen.de and are subject to approval by the KORA Board. Analysis codes are available from the authors upon reasonable request."

Reviewers' comments:

Reviewer's Responses to Questions

**Comments to the Author**

1. Is the manuscript technically sound, and do the data support the conclusions?

Reviewer #1: Yes

Reviewer #2: Yes

2. Has the statistical analysis been performed appropriately and rigorously? 

Reviewer #1: Yes

Reviewer #2: Yes

3. Have the authors made all data underlying the findings in their manuscript fully available?

Reviewer #1: Yes

Reviewer #2: No

4. Is the manuscript presented in an intelligible fashion and written in standard English?

Reviewer #1: Yes

Reviewer #2: Yes

5. Review Comments to the Author

Reviewer #1: General comments:

This is a very interesting manuscript dealing with a topic of importance in the field of cardiology.

Specific comments:

Methods, Study sample, last paragraph, Page 5: the authors should state the population that was accounted for the analysis e.g. participants with the 3 exams and endpoint at follow-up, and from there indicating the excluded participants.

Methods, association of trajectory clusters with cardiometabolic events, Page 8: the authors do not state the population on which they performed kaplan meïer methods which may be different than participants with 3 exams.

Methods, association of trajectory clusters with cardiometabolic events, Page 8: in a more general aspect definition of clusters of trajectory should be made clear for each analysis e.g. prevalent cardiometabolic events, Kaplan-Meïer analysis and incident cardiometabolic events. This is not clear in the methods.

Results, Study sample, Page 8: Authors may also give antihypertensive consumption in the analysed population during the exams, along with decreasing mean levels of blood pressure.

Results, Characterisation of longitunal trajectory clusters, Page 9: a better description of the clusters would be nice to have with a charaterisation of each (what is it exactly?) instead of only interpreting values comparatively. The high risk cluster seem to have a different behavior for DBP, BMI, HBA1c, Total and LDL cholesterol both in emn and women and this should be mentionned. Is it also possible to validate the cluster against established and published severity values of the different risk factors?

Results: I as not able to find results of the Kaplan Meïer except sensitivity analyses in appendix.

Reviewer #2: In this manuscript, the authors explored the effects of joint longitudinal trajectories of multiple risk factors in cardiometabolic events from the population-based cohort. The single time-point measure of metabolic parameters provides limit evidences in predicting cardiovascular events, while the assessment of cumulative effects of metabolic disorders are prominent in predicting the clinical outcomes. This manuscript is well written and the conclusion is convincing. While there are some concerns needed to be solved before the publication.

1. The “nine cardiometabolic risk factors” in abstract should be described.

2. The population were divided into three cluster according to Calinski-Harabasz criterion. Therefore, the clinical characteristics should be provided and compared among these clusters.

3. The data were analyzed in the sex-specific trajectory clusters, while the difference between the cardiometabolic events were solely attributed to sex hormones in the Discussion.

4. The discussion should be more precise.

6. PLOS authors have the option to publish the peer review history of their article (what does this mean?). If published, this will include your full peer review and any attached files.

Reviewer #1: **Yes: **Michel Vaillant, PhD

Reviewer #2: No

---

## [Author Response · Author response to Decision Letter 0]

21 Feb 2024

Reviewer 1

This is a very interesting manuscript dealing with a topic of importance in the field of cardiology.

Answer: We thank the reviewer for the positive evaluation of our work and for the detailed comments. We have addressed each comment and made the respective changes in the manuscript and feel the manuscript has improved.

Comment: 1) Methods, Study sample, last paragraph, Page 5: the authors should state the population that was accounted for the analysis e.g. participants with the 3 exams and endpoint at follow-up, and from there indicating the excluded participants.

Answer: We thank the reviewer for raising this point. We rephrased the last paragraph in the methods section “study sample“ and now start with the sample soze of the final analytical sample. 

Comment: 2) Methods, association of trajectory clusters with cardiometabolic events, Page 8: the authors do not state the population on which they performed kaplan meïer methods which may be different than participants with 3 exams.

Answer: We thank the reviewer for this observation. For the Kaplan-Meier analyses, we used the same sample as for the cluster analysis, except for the exclusion of n=72 prevalent cases at baseline. We have now clarified this in the respective paragraph in the Methods, section “Association of trajectory clusters with cardiometabolic events”. This sample comprised all participants with complete data at all three exams. We did not consider mortality as an event between Exam 1 and Exam 3, as described in the Methods. 

Comment: 3) Methods, association of trajectory clusters with cardiometabolic events, Page 8: in a more general aspect definition of clusters of trajectory should be made clear for each analysis e.g. prevalent cardiometabolic events, Kaplan-Meïer analysis and incident cardiometabolic events. This is not clear in the methods.

Answer: We apologize for the lack of clarity. We have now included a diagram showing the structure and sequence of the analyses for cluster derivation, KM curves, and the regression models in Supplementary Figure S2, and added an overview paragraph to the Statistical Analysis section. In addition, we expanded the legend of Table 2 and Figures 4, S4 and S6 to include information on the sample size used. Briefly, we used the final sample of n = 976 men and n = 1,004 women to derive clusters. For the visualization of event-free survival by KM curves and for the quantification of prevalent events at Exam 3 by logistic regression models, we excluded n = 72 participants with prevalent events at Exam 1, resulting in a sample of n = 926 men and n = 982 women. For quantification of incident events at Exam 3 by logistic regression models, we excluded n = 292 participants with prevalent events at Exam 3, resulting in a sample of n = 806 men and n = 892 women. We have added a description of this in the Methods section, “Association of trajectory clusters with cardiometabolic events”.

Comment: 4) Results, Study sample, Page 8: Authors may also give antihypertensive consumption in the analysed population during the exams, along with decreasing mean levels of blood pressure.

Answer: We thank the reviewer for this observation. We have added this information to the corresponding description of the blood pressure reductions in the section Results, “Study sample”.

Comment: 5) Results, Characterisation of longitunal trajectory clusters, Page 9: a better description of the clusters would be nice to have with a charaterisation of each (what is it exactly?) instead of only interpreting values comparatively. The high risk cluster seem to have a different behavior for DBP, BMI, HBA1c, Total and LDL cholesterol both in emn and women and this should be mentionned. Is it also possible to validate the cluster against established and published severity values of the different risk factors?

Answer: The reviewer mentions an important point, also observed by Reviewer #2, Comment 2). We have now substantially rewritten the results section on “Characterization of longitudinal trajectory clusters” to better describe the individual clusters for men and women. We now also include a more through description of sex-specific risk factor patterns. Furthermore, we have added two Supplementary Tables (S5 and S6 Table) showing mean risk factor values according to cluster membership, corresponding to Figures 1 and 2 (line charts). Furthermore, we have incorporated the relation with established thresholds for disease diagnoses, e.g., hypertension, obesity, dyslipidemia in the section Results, “Characterization of longitudinal trajectory clusters”.

Comment: 6) Results: I as not able to find results of the Kaplan Meïer except sensitivity analyses in appendix.

Answer: The KM curves were supposed to be Figure 4 in the main manuscript. We apologize that this was not visible in the first submission, we hope this issue is now fixed.

Reviewer 2

Comment: In this manuscript, the authors explored the effects of joint longitudinal trajectories of multiple risk factors in cardiometabolic events from the population-based cohort. The single time-point measure of metabolic parameters provides limit evidences in predicting cardiovascular events, while the assessment of cumulative effects of metabolic disorders are prominent in predicting the clinical outcomes. This manuscript is well written and the conclusion is convincing. While there are some concerns needed to be solved before the publication.

Answer: We thank the reviewer for the positive evaluation of our work and for the detailed comments. We have addressed each comment and made the respective changes in the manuscript and feel the manuscript has improved.

Comment: 1. The “nine cardiometabolic risk factors” in abstract should be described.

Answer: We have added the respective information to the abstract. 

Comment: 2. The population were divided into three cluster according to Calinski-Harabasz criterion. Therefore, the clinical characteristics should be provided and compared among these clusters.

Answer: The reviewer mentions an important point, also observed by Reviewer #1, Comment 5). We have now substantially re-worked the results section on “Characterization of longitudinal trajectory clusters” to better describe the individual clusters for men and women. Furthermore, we have added two Supplementary Tables (S5 and S6 Table) with the mean risk factor values according to cluster membership, corresponding to Figures 1 and 2 (line charts). We related the mean risk factor levels to established thresholds for disease diagnoses, e.g., hypertension, obesity, dyslipidemia.

Comment: 3. The data were analyzed in the sex-specific trajectory clusters, while the difference between the cardiometabolic events were solely attributed to sex hormones in the Discussion.

Answer: We thank the reviewer for raising this point. We have now added a more thorough discussion of potential reasons for sex-specific differences, including differences in treatment, therapy adherence, underrepresentation in clinical trials, etc., in the respective discussion section. 

Comment: 4. The discussion should be more precise.

Answer: We have substantially re-worked the discussion section. Besides incorporating the points mentioned above, we have rephrased, summarized, and shortened parts of the discussion to make it more precise.

---

## [Decision Letter · Decision Letter 1]

7 Mar 2024

Clusters of longitudinal risk profile trajectories are associated with cardiometabolic diseases: results from the population-based KORA cohort

PONE-D-23-38330R1

Dear Dr. Rospleszcz,

We’re pleased to inform you that your manuscript has been judged scientifically suitable for publication and will be formally accepted for publication once it meets all outstanding technical requirements.

Kind regards,

Peng Gao, Ph.D.

Academic Editor

PLOS ONE

Additional Editor Comments (optional):

Reviewers' comments:

Reviewer's Responses to Questions

**Comments to the Author**

1. If the authors have adequately addressed your comments raised in a previous round of review and you feel that this manuscript is now acceptable for publication, you may indicate that here to bypass the “Comments to the Author” section, enter your conflict of interest statement in the “Confidential to Editor” section, and submit your "Accept" recommendation.

Reviewer #1: All comments have been addressed

Reviewer #2: All comments have been addressed

2. Is the manuscript technically sound, and do the data support the conclusions?

Reviewer #1: Yes

Reviewer #2: Yes

3. Has the statistical analysis been performed appropriately and rigorously? 

Reviewer #1: Yes

Reviewer #2: Yes

4. Have the authors made all data underlying the findings in their manuscript fully available?

Reviewer #1: No

Reviewer #2: Yes

5. Is the manuscript presented in an intelligible fashion and written in standard English?

Reviewer #1: Yes

Reviewer #2: Yes

6. Review Comments to the Author

Reviewer #1: All previous comments were addressed. The added precisions and supplementary materials are sufficient for the reader to fully understand and follow the work in the article. There is no further comment.

Reviewer #2: The maniscript is well-written and the results are further interpreted. The concerns have been clearly addressed and i am statisficed with this revised version.

7. PLOS authors have the option to publish the peer review history of their article (what does this mean?). If published, this will include your full peer review and any attached files.

Reviewer #1: **Yes: **Michel Vaillant

Reviewer #2: No

---

## [Editor Report · Acceptance letter]

15 Mar 2024

PONE-D-23-38330R1 

PLOS ONE

Dear Dr. Rospleszcz, 

I'm pleased to inform you that your manuscript has been deemed suitable for publication in PLOS ONE. Congratulations! Your manuscript is now being handed over to our production team.

Kind regards, 

on behalf of

Professor Peng Gao 

Academic Editor

PLOS ONE